# Colorectal Cancer among Resettlers from the Former Soviet Union and in the General German Population: Clinical and Pathological Characteristics and Trends

**DOI:** 10.3390/ijerph18094547

**Published:** 2021-04-25

**Authors:** Melani Ratih Mahanani, Simone Kaucher, Hiltraud Kajüter, Bernd Holleczek, Heiko Becher, Volker Winkler

**Affiliations:** 1Epidemiology of Transition, Heidelberg Institute of Global Health, University Hospital Heidelberg, 69120 Heidelberg, Germany; melani.mahanani@uni-heidelberg.de (M.R.M.); simone.kaucher@uni-heidelberg.de (S.K.); 2Cancer Registry, 44801 Bochum, Germany; Hiltraud.Kajueter@krebsregister.nrw.de; 3Saarland Cancer Registry, 66119 Saarbrücken, Germany; b.holleczek@krebsregister.saarland.de; 4Institute for Medical Biometry and Epidemiology, University Medical Center Hamburg-Eppendorf, 20246 Hamburg, Germany; h.becher@uke.de

**Keywords:** incidence, colorectal cancer, young-onset, clinical characteristics, pathological characteristics, migrants, Former Soviet Union, Germany

## Abstract

This study examined time trends and clinical and pathological characteristics of colorectal cancer (CRC) among ethnic German migrants from the Former Soviet Union (resettlers) and the general German population. Incidence data from two population-based cancer registries were used to analyze CRC as age-standardized rates (ASRs) over time. The respective general populations and resettler cohorts were used to calculate standardized incidence ratios (SIRs) by time-period (before and after the introduction of screening colonoscopy in 2002), tumor location, histologic type, grade, and stage at diagnosis. Additionally, SIRs were modeled with Poisson regression to depict time trends. During the study period from 1990 to 2013, the general populations showed a yearly increase of ASR, but for age above 55, truncated ASR started to decline after 2002. Among resettlers, 229 CRC cases were observed, resulting in a lowered incidence for all clinical and pathological characteristics compared to the general population (overall SIR: 0.78, 95% CI 0.68–0.89). Regression analysis revealed an increasing SIR trend after 2002. Population-wide CRC incidence decreases after the introduction of screening colonoscopy. In contrast the lowered CRC incidence among resettlers is attenuating to the general population after 2002, suggesting that resettlers do not benefit equally from screening colonoscopy.

## 1. Introduction

The term colorectal cancer (CRC) summarizes malignancies of the colon and the rectum. In 2016, its median age of diagnosis was 76 and 72 years among German women and men, respectively [1]. The incidence of CRC started to decrease after the introduction of colon cancer screening by colonoscopy in 2002 [2]. Recently, rising CRC incidence among adolescents and young adults has attracted increased attention [3,4]. It is known that these so-called young-onset CRC tumors present distinctive clinical and pathological characteristics with lower survival compared to non-young-onset CRC cases [4,5,6]. An increasing prevalence of well-known CRC risk factors, such as alcohol consumption [7], red meat intake, low physical activity, cigarette smoking [8], obesity [9,10], and diabetes mellitus [11] are discussed as the main reasons. Additionally, the recommended screening age may influence the observed differences between young-onset and non-young-onset CRC, respectively [12,13].

Colonoscopy is a secondary prevention method that is generally offered to populations with increased age, family history of CRC syndromes, and chronic inflammatory bowel diseases [4,14]. Unlike other cancer screening methods aiming for early diagnosis to improve patients’ outcomes, colonoscopy additionally allows removing precancerous lesions during the examination [15]. In Germany, statutory health insurance (SHI) has offered fully covered colonoscopies to people aged 55 years and above since 2002, as an alternative to the fecal occult blood test, which has been offered since 1977 [16]. In 2019, Germany lowered the recommended age for screening colonoscopies for men to 50 years, while it remained unchanged for women [17].

Ethnic German resettlers from the Former Soviet Union are the second-largest migrant group in Germany, with about 2.5 million people immigrating in large numbers in the early 1990s after the fall of the iron curtain [18]. They receive German citizenship upon arrival and are entitled to fully utilize the German healthcare system [19,20]. Focusing on cancer incidence and mortality compared to the general population of Germany, Kaucher et al. found that incidence and mortality of colorectal (both sexes), lung (women), prostate, and female breast cancer were lower among resettlers [21].

Considering the discussion about the increasing incidence of CRC at younger ages and the offer of screening colonoscopy, this study aims to explore the incidence of CRC and its temporal trends and to compare clinical and pathological characteristics of CRC cases between resettlers and the general German population.

## 2. Materials and Methods

We used data of two resettler cohorts, one in the administrative district of Münster (North Rhine-Westphalian) and another one in the federal state of the Saarland with the observation periods 1994 to 2013 and 1990 to 2009, respectively. The combined cohort comprised 51,311 resettlers (Saarland: 18,619; Münster: 32,692), who immigrated between 1990 and 2001 (Münster) and between 1990 and 2005 (Saarland). More details on the study population and the follow-up procedures can be found elsewhere [21,22]. The study protocol was approved by the Ethics Committee of the Medical Faculty, University Hospital Heidelberg [22].

In brief, the vital status of resettlers was derived from local population registries through record linkage or manually. The accumulated person-time was estimated for each sex, 5-year age group (up to 85+), and calendar year. For the general populations of the Saarland and Münster, person-time was ascertained from the mid-year populations provided by the federal statistic office of the Saarland and the federal cancer registry of North Rhine-Westphalian, respectively. Both population-based cancer registries provided data on CRC cases in the respective general population and the respective resettler cohort through record linkage. Incidence data included date of diagnosis, age at diagnosis, sex, tumor location (International Classification of Diseases 10th Revision (ICD-10)), tumor morphology (International Classification of Diseases for Oncology third revision (ICD O-3)), tumor grade, and stage at diagnosis. We restricted all analysis to histologically confirmed primary CRC cases (ICD-10 C18-C20) and categorized patients according to age (<55 years: young-onset CRC, ≥55 years: non-young-onset CRC).

The condensed stage at diagnosis coding system developed by the European Network of Cancer Registries was used to categorize tumors into a local, advanced, or unknown stage based on the status of lymph nodes (N) and the existence of metastasis (M) [23], due to expected missing values in population-wide registry data with varying versions of the TNM classification. Furthermore, we combined well and moderately differentiated tumors (grades 1 and 2) as low grade and poorly differentiated and undifferentiated ones (grades 3 and 4) as high grade. For three-year calendar periods, we calculated truncated age-standardized incidence rates (ASRs) separately for young-onset and non-young-onset CRC using the 1976 European standard population [24].

Afterward, we modeled the truncated ASR over time by first estimating age-specific rates with Poisson regression using the number of observed cases as the dependent variable and the log of the mid-year population P as the offset. For young-onset incidence Y, calendar year T from 1990 to 2013 (continuous, coded from 0 to 23, starting in 1990) and age group A (categorical, 5-year age groups) were used as covariables (see formula 1).

For non-young-onset CRC incidence *Z*, an interrupted time-series approach [25] was used to detect changes due to colonoscopy screening introduced in 2002. In addition to calendar year *T* and age group *A*, “colonoscopy” *X* (binary; 0: years 1990–2001, 1: years 2002–2013) and the interaction term between the calendar year and “colonoscopy” were used as covariables (see Formula (2)). Using the modeled yearly age-specific rates *Y* and *Z* from Formulas (1) and (2), we calculated the ASR shown in Figure 1:(1)$$\log\left(Y_{t,\;a}\right)=\log\left(P\right)+\beta_{0}+\beta_{1}T+\beta_{2}^{\prime }A$$
(2)$$\log\left(Z_{t,\;a}\right)=\log\left(P\right)+\beta_{0}+\beta_{1}T+\beta_{2}^{\prime }A+\beta_{3}X_{t}+\beta_{4}TX_{t}$$

In the next step, we calculated standardized incidence ratios (SIRs) to compare observed CRC cases among resettlers to expected numbers in the respective host population using sex, age group, and calendar year-specific rates. SIRs were computed with exact 95% confidence intervals (95% CI) for all cases and before and after the introduction of colonoscopy in 2002, as well as for tumor location, grading, histology, and condensed stage at diagnosis.

We also modeled sex-specific SIRs with Poisson regression for young-onset and non-young onset cases separately using the number of observed cases among resettlers as the dependent variable and the log of the expected cases *E* as the offset. For young-onset cases *U*, we used sex *S* (binary; 0: male, 1: female) and calendar year *T* as covariables (see Formula (3)). For non-young-onset cases V, the model was again extended with the covariables colonoscopy *X* and the interaction term of the calendar year and colonoscopy (see Formula (4)):(3)$$\log\left(U_{t,\;s}\right)=\log\left(E\right)+\beta_{0}+\beta_{1}T+\beta_{2}S$$
(4)$$\log\left(V_{t,\;s}\right)=\log\left(E\right)+\beta_{0}+\beta_{1}T+\beta_{2}S+\beta_{3}X_{t}+\beta_{4}TX_{t}$$

In all Poisson models, standard errors were controlled for overdispersion [26]. Statistical analyses were performed using Stata/IC 15.1 for Windows (64-bit x86-64) Revision 21 November 2017 (StataCorp LLC, 4905 Lakeway Drive, College Station, TX 77845, USA).

## 3. Results

Between 1990 and 2013, the combined resettler cohorts comprised 666,899 person-years and 238 diagnoses with primary colorectal cancer, of which 229 (96.2%) cases were histologically confirmed. In the host population, 48,980 (88.7%) CRC cases were histologically confirmed. Demographic characteristics of patients, as well as clinical and pathological features of the included tumors among the general population and the resettlers, are summarized in Table 1. Notably, there were more young-onset CRC cases among resettlers (22.3% vs. 10.0%). Rectal cancer was more frequently diagnosed for the general population, while among resettlers, CRC was more likely in the left colon. In both groups, most CRC cases were of other adenocarcinoma subtypes, low grade, and localized tumors.

Separated by age at onset, Figure 1 illustrates the observed and the modeled truncated ASR of the general populations. The underlying Poisson regression coefficients can be found in Appendix A.

Table 2 presents results of the SIR analyses of CRC among resettlers compared to the general population. Overall, the SIR was lower among resettlers in both cohorts and for both sexes. Resettlers showed a lower incidence of CRC according to all clinical and pathological characteristics.

Modeled SIRs for CRC among resettlers are shown in Figure 2. Among non-young-onset CRC, an increasing SIR could be observed after the introduction of screening colonoscopy in 2002. Corresponding Poisson regression coefficients can be found in Appendix A.

## 4. Discussion

This study confirmed a declining incidence in the population for which screening colonoscopy is offered in Germany. Among resettlers, the CRC incidence in older age groups was lower compared to the general population; however, for young-onset CRC, there was no difference. The lower incidence of resettlers was increasing and, therefore, attenuating to the general population after the introduction of screening colonoscopy. With respect to clinical and pathological characteristics, there were hardly any differences to the general population except for a higher frequency of left colon tumors among resettlers.

The observed increase of CRC incidence among young individuals is consistent with a number of studies from different western countries [10,27,28,29]. This increase may to some extent be attributed to modifiable risk factors, such as obesity and physical inactivity [30,31]. Additionally, low awareness of young-onset CRC among both patients and physicians and that it also occurs in those who are not subjected to family history or apparent risk factors [32] might contribute.

Left-sided CRC diagnosis is associated with rectal bleeding and changes in bowel habits [33], which may generally lead to delayed diagnoses. Additionally, a previous study suggested that male resettlers were more likely to be diagnosed with advanced tumors when looking at the most frequent cancer-sites combined (stomach, colorectal, lung, breast, and prostate cancer) [21]. However, this study does not show delayed diagnoses for CRC among resettlers. In contrast, the general German population presented a higher incidence of mucinous adenocarcinoma, which is associated with poorer clinical and pathological characteristics, such as higher grade and advanced stage at diagnosis, leading to lower survival compared to other CRC types [34].

A possible explanation for the attenuating incidence between resettlers and the general population is that risk behaviors and lifestyle adjustments to the host population are likely among migrants, as well as improved screening and diagnostic accessibility [21,35]. Resettlers may gradually adjust their lifestyle and dietary habits due to greater availability and selection of food [21]. Another explanation for the attenuating incidence might be the overtime constant CRC incidence rate among resettlers, suggesting that resettlers do not benefit from screening colonoscopy equally to the German population, which experiences decreasing rates. If resettlers do not use screening colonoscopy, they also do not benefit from the possibility to remove precancerous lesions, which may result in a higher incidence of CRC. However, the constant CRC incidence rate among resettlers (analysis not shown) might also be explained by the limited number of observations.

Our study is the first population-based study looking at time trends and clinical and pathological characteristics of young-onset and non-young-onset CRC among resettlers from the Former Soviet Union compared to Germany’s general population. It needs to be stated that the analysis relies only on secondary data without information on individual risk factors, such as lifestyle, family history of CRC, etc. Furthermore, the dataset was restricted to histologically confirmed CRC cases leavening out 11.0% (young-onset: 5.2%; non-young-onset: 11.9%) of all reported CRC cases. However, there was no time trend concerning histological confirmation, and the fraction of confirmed cases was close to or above 90% except for the years 1994 to 1996 when only about 70% of all cases were histologically confirmed. Therefore, the restriction to histologically confirmed cases does not introduce bias onto the time trend analysis of the general population. Concerning the resettler cohorts, selection bias was unlikely since all ethnic Germans were invited to migrate to Germany, and during the immigration process, they were allocated quasi-randomly to their first area of residence [22]. Due to data protection concerns, neither information on the date of immigration nor an individual mortality follow-up among individuals of the Münster cohort was available, which prevented us from analyzing the incidence among resettlers concerning lengths of stay in Germany. However, since most resettlers migrated to Germany in the first half of the 1990s, calendar time is highly correlated with length of stay. It should also be mentioned that the person-time of the Münster cohort had to be estimated due to an incomplete follow-up [36].

## 5. Conclusions

Similar to other countries, Germany is encountering a decreasing CRC incidence in the population eligible for screening colonoscopy. CRC incidence among ethnic German migrants from the Former Soviet Union is lower but continuously attenuates to the general population. This might hint towards less screening participation among resettlers, which may lead to increasing CRC incidence. However, the clinical and pathological characteristics of the resettler’s tumor conditions were hardly different from the general population.

## Figures and Tables

**Figure 1 ijerph-18-04547-f001:**
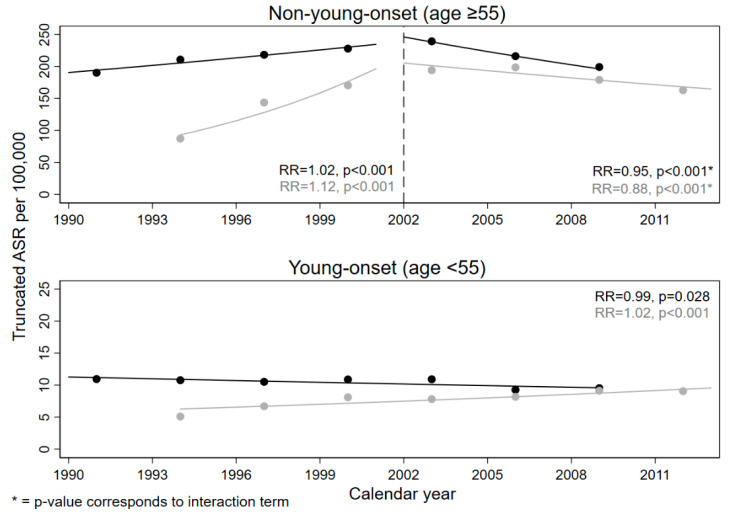
Observed and modeled young-onset and non-young-onset truncated ASRs of colorectal cancer incidence for the general population from 1990 to 2013; the rate ratio (RR) corresponds to the modeled calendar year effect (see Appendix A); the dashed line indicates the introduction of screening colonoscopy; black represents the Saarland population, gray the Münster population.

**Figure 2 ijerph-18-04547-f002:**
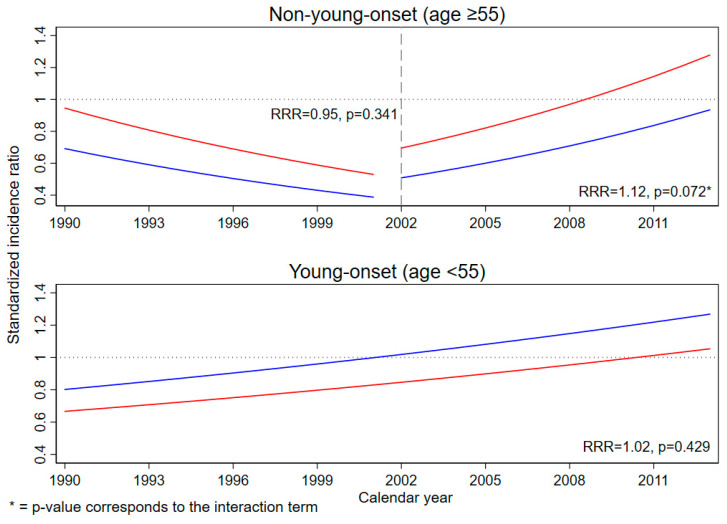
Modeled standardized incidence ratios for colorectal cancer among resettlers compared to the direct host populations using Poisson regression from 1990 to 2013; the relative SIR change (RRR) corresponds to the modeled calendar year effect (see Appendix A); the dashed line indicates the year of screening colonoscopy introduction; the blue line represents men, red line women.

**Table 1 ijerph-18-04547-t001:** Demographic characteristics and clinical and pathological characteristics of CRC among resettlers and in the general population (Saarland, 1990–2009 and Münster, 1994–2013).

Characteristics		General Population		Resettler	
		N	%	N	%
**Total**		48,980	100.0	229	100.0
**Region**	Saarland	17,405	35.5	76	33.2
	Münster	31,575	64.5	153	66.8
**Time period**	1990–2001	19,466	39.7	52	22.7
	2002–2013 (colonoscopy)	29,514	60.3	177	77.3
**Young-onset**	yes (age < 55)	4906	10.0	51	22.3
	no (age ≥ 55)	44,074	90.0	178	77.7
**Sex**	Female	25,349	51.8	106	46.3
	Male	23,631	48.2	123	53.7
**Anatomic location**	Right colon	13,123	26.8	55	24.0
	Left colon	12,756	26.0	83	36.3
	Rectum	15,810	32.3	66	28.8
	Other/unknown	7291	14.9	25	10.9
**Histologic Type**	Mucinous adenocarcinoma	8463	17.3	34	14.9
	Signet-ring cell carcinoma	311	0.6	1	0.4
	Other adenocarcinoma subtypes	36,344	74.2	183	79.9
	Other/unknown	3862	7.9	11	4.8
**Tumor grade**	Low	34,663	70.8	174	76.0
	High	10,499	21.4	44	19.2
	Unknown	3818	7.8	11	4.8
**Tumor stage**	Local	18,469	37.7	93	40.6
	Advanced	17,404	35.5	85	37.1
	Unknown	13,107	26.8	51	22.3

**Table 2 ijerph-18-04547-t002:** Standardized incidence ratios of resettlers compared to the general Saarland (1990–2009) and Münster (1994–2013) population with exact 95% confidence intervals.

Characteristics		Total		Saarland		Münster	
		Obs.	SIR (95% CI)	Obs.	SIR (95% CI)	Obs.	SIR (95% CI)
**Total (1990–2013)**		229	**0.78 (0.68–0.89)**	76	**0.73 (0.57–0.91)**	153	**0.81 (0.68–0.94)**
**Time period**	1990–2001	52	**0.61 (0.46–0.80)**	31	0.72 (0.49–1.02)	21	**0.50 (0.31–0.76)**
	2002–2013 (colonoscopy)	177	**0.85 (0.73–0.98)**	45	**0.74 (0.54–0.98)**	132	0.89 (0.75–1.06)
**Young-onset**	Yes (age < 55)	51	0.99 (0.74–1.31)	14	0.80 (0.44–1.34)	37	1.10 (0.77–1.51)
	No (age ≥ 55)	178	**0.73 (0.63–0.85)**	62	**0.72 (0.55–0.92)**	116	**0.74 (0.61–0.89)**
**Sex**	Female	123	0.85 (0.71–1.02)	47	0.97 (0.71–1.29)	76	**0.80 (0.63–0.99)**
	Male	106	**0.70 (0.58–0.85)**	29	**0.52 (0.35–0.75)**	77	0.81 (0.64–1.02)
**Anatomical location**	Right colon	55	**0.69 (0.52–0.91)**	15	0.65 (0.36–1.06)	40	**0.72 (0.51–0.98)**
	Left colon	83	1.08 (0.86–1.33)	29	1.08 (0.73–1.55)	54	1.07 (0.80–1.39)
	Rectum	66	**0.68 (0.53–0.87)**	16	**0.45 (0.26–0.73)**	50	0.82 (0.61–1.08)
	Others (incl. % unknown)	25 (20)	**0.60 (0.39–0.89)**	16 (31.3)	0.86 (0.49–1.40)	9 (0)	**0.39 (0.18–0.75)**
**Histologic type**	Mucinous adenocarcinoma	34	**0.67 (0.47–0.94)**	14	0.82 (0.45–1.38)	20	**0.60 (0.36–0.92)**
	Signet-ring cell carcinoma	1	0.45 (0.01–2.49)	1	1.13 (0.03–6.31)	0	0.00 (0.00–2.73)
	Other adenocarcinomas	183	**0.83 (0.71–0.96)**	59	**0.73 (0.56–0.94)**	124	0.88 (0.74–1.05)
	Others	11	**0.53 (0.26–0.95)**	2	0.35 (0.04–1.26)	9	0.60 (0.27–1.14)
**Tumor grade**	Low grade	174	**0.83 (0.71–0.96)**	61	0.82 (0.62–1.05)	113	**0.83 (0.68–0.99)**
	High grade	44	**0.68 (0.50–0.92)**	13	**0.58 (0.31–0.98)**	31	0.74 (0.50–1.05)
	Unknown	11	0.57 (0.29–1.03)	2	0.29 (0.03–1.04)	9	0.73 (0.34–1.40)
**Tumor stage**	Local stage	93	0.82 (0.66–1.01)	26	**0.67 (0.44–0.98)**	67	0.90 (0.70–1.15)
	Advanced stage	85	**0.79 (0.63–0.98)**	32	0.86 (0.59–1.22)	53	**0.76 (0.57–0.99)**
	Unknown	39	**0.64 (0.45–0.87)**	18	**0.49 (0.25–0.88)**	28	0.72 (0.48–1.05)

SIR, standardized incidence ratio; Obs., number of observations; CI, confidence interval. Significant results are bolded.

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
