# Peer review of "Colorectal Cancer among Resettlers from the Former Soviet Union and in the General German Population: Clinical and Pathological Characteristics and Trends"

_ijerph, 2021, doi:10.3390/ijerph18094547_

Round 1

Reviewer 1 Report

The manuscript entitled „Colorectal cancer among resettlers from the Former Soviet Union and in the general German population: clinical and pathological characteristics and trends” presents interesting issue but the manuscript requires major corrections.

Major:

The presented manuscript seems rather to be only a draft of the future manuscript.

Authors should get familar with the manuscripts published in IJERPH journal to correct their writing style and content of the manuscript to be in agreement with the general academic standards and present a proper scientific regime.

General:

Autors should prepare their manuscript according to instructions for authors

Abstract:

Authors should present brief justification of the study.

Authors should present the aim of the study (e.g. „The aim of the study was…”), not only what was done.

Authors should briefly present the registers that were used.

Introduction:

Authors should prepare this section not only to be interesting for German readers, but to be interesting for international readers. If Authors prepare their manuscript only for their national readers, they should publish it in some national journal. So, Authors should present here a proper justification of their study, not based only on German data but international ones.

Authors should properly justify need to conduct such study in a population of resettlers from the Former Soviet Union – what is the general characteristics and history of this population, why was this population so specific to suppose that the results of the study may be of the value, etc. (the justification can not be based only on a numer of resettlers in Germany – if Authors want their manuscript to be interesting not only for national, but also for international readers)

Materials and Methods:
As the confidential data were used, Authors should provide information about ethical commitee agreement

This section needs really major corrections.

Authors should present their methodology in details while justifying their actions – e.g. „we combined well and moderately differentiated tumors (grades 1 and 2) as low grade” – why AUthors did it – such decisions should be based on literature (this is only one example while such corrections should be made in the whole section)

It seems that Authors did not conduct proper statistical analysis (Table 1) – it should be conducted and presented within the Materials and Methods Section

Resuls:

It seems that Authors did not conduct proper statistical analysis (Table 1), as the samples were not properly compared. However Authors comment their results (that the share in one population was higher than in the other) – they can not do it if they did not conduct statistical analysis – it should be conducted and the results should be presented (p-Values)

Discussion:

The discussion should be broadened – Authos can not only describe their results, but they must compare tchem with the results of other authors ( not only their own studies), especially those conducted in other countries. As infdicated above, if Authors prepare their manuscript only for their national readers, they should publish it in some national journal.

Authors should discuss the limitations of their study.

Conclusions:

Authors should present here only the conclusions based on the conducted study – e.g. if Authors did not conduct the analysis of international data, they can not conclude about international similarities/ differneces.

References:

Authors should reduce the numer of self-citations (5 of them)

Reviewer 2 Report

This is a robust study on the incidence of colorectal cancer in the German population in general and the German population from the former Soviet Union.
The data obtained are explained in an objective and direct manner, and the discussion and conclusion are based on the respective findings. Anyway, that the study adds important and pertinent knowledge in the area, as proposed by the author.
My question is about the ethical aspects. The authors did not indicate in their methodology approval of the study by an ethics committee relevant to the development of the study. If approved, please add this information to the methodology.

Reviewer 3 Report

This is an interesting study that aimed to investigate the incidence of colorectal cancer (CRC) and its temporal trends and to compare clinical and pathological characteristics of colorectal cancer cases between ethnic German migrants from the Former Soviet Union and the general German population. This study showed some strength points but also weaknesses. Overall, the study has been well carried out with a good experimental design. The two combined cohorts of patients comprised 51,311 resettlers, which represented a highly representative sample. Statistical models were accurate and well developed. The results are presented clearly through tables and figures, which included all the relevant information. The authors stated that their study is “the first population-based study looking at time trends and clinical and  pathological characteristics of young-onset and non-young-onset CRC among resettlers  from the Former Soviet Union in comparison to Germany’s general population”. From a look to the literature this seems to be correct, thus confirming the novelty of the research.

With regard to the weaknesses, the Authors highlighted a number of limitations, such as the lack of data on information on individual risk factors such as lifestyle, family history of CRC, etc. Moreover, the analysis has been performed only on histologically confirmed CRC cases leavening out 11.0%  of all reported CRC cases. Nonetheless, the Authors reported that the restriction to histologically confirmed cases  did not introduce bias onto the time trend analysis of the general population and discussed the reasons in the text.

The Authors discussed only the role of screening colonoscopy. It would be interesting to discuss the role and the importance of other screening tools, such as fecal occult blood test (FOBT), biomarkers, liquid biopsy etc.

In conclusion, this is a well conducted study and the findings are novel and interesting. However, more studies are needed to investigate the differences between the two populations (resettlers vs general population).

Round 2

Reviewer 1 Report

The manuscript entitled „Colorectal cancer among resettlers from the Former Soviet Union and in the general German population: clinical and pathological characteristics and trends” presents interesting issue but the manuscript requires major corrections.

Major:

The presented manuscript seems rather to be only a draft of the future manuscript.

Authors should get familar with the manuscripts published in IJERPH journal to correct their writing style and content of the manuscript to be in agreement with the general academic standards and present a proper scientific regime.

General:

Autors should prepare their manuscript according to instructions for authors

Abstract:

Authors should present brief justification of the study.

Authors should present the aim of the study (e.g. „The aim of the study was…”), not only what was done.

Authors should briefly present the registers that were used.

Introduction:

Authors should prepare this section not only to be interesting for German readers, but to be interesting for international readers. If Authors prepare their manuscript only for their national readers, they should publish it in some national journal. So, Authors should present here a proper justification of their study, not based only on German data but international ones.

Authors should properly justify need to conduct such study in a population of resettlers from the Former Soviet Union – what is the general characteristics and history of this population, why was this population so specific to suppose that the results of the study may be of the value, etc. (the justification can not be based only on a numer of resettlers in Germany – if Authors want their manuscript to be interesting not only for national, but also for international readers)

Materials and Methods:

This section needs really major corrections.

Authors should present their methodology in details while justifying their actions – e.g. „we combined well and moderately differentiated tumors (grades 1 and 2) as low grade” – why AUthors did it – such decisions should be based on literature (this is only one example while such corrections should be made in the whole section)

It seems that Authors did not conduct proper statistical analysis (Table 1) – it should be conducted and presented within the Materials and Methods Section

Resuls:

It seems that Authors did not conduct proper statistical analysis (Table 1), as the samples were not properly compared. However Authors comment their results (that the share in one population was higher than in the other) – they can not do it if they did not conduct statistical analysis – it should be conducted and the results should be presented (p-Values)

Discussion:

The discussion should be broadened – Authos can not only describe their results, but they must compare tchem with the results of other authors ( not only their own studies), especially those conducted in other countries. As infdicated above, if Authors prepare their manuscript only for their national readers, they should publish it in some national journal.

Authors should discuss the limitations of their study.

Conclusions:

Authors should present here only the conclusions based on the conducted study – e.g. if Authors did not conduct the analysis of international data, they can not conclude about international similarities/ differneces.

References:

Authors should reduce the numer of self-citations (5 of them)
